# Design and Analysis of a Contact Piezo Microphone for Recording Tracheal Breathing Sounds

**DOI:** 10.3390/s24175511

**Published:** 2024-08-26

**Authors:** Walid Ashraf, Zahra Moussavi

**Affiliations:** Biomedical Engineering Program, University of Manitoba, Winnipeg, MB R3T 2N2, Canada

**Keywords:** tracheal breathing sounds, respiratory disorders diagnosis, contact microphone, surface microphone

## Abstract

Analysis of tracheal breathing sounds (TBS) is a significant area of study in medical diagnostics and monitoring for respiratory diseases and obstructive sleep apnea (OSA). Recorded at the suprasternal notch, TBS can provide detailed insights into the respiratory system’s functioning and health. This method has been particularly useful for non-invasive assessments and is used in various clinical settings, such as OSA, asthma, respiratory infectious diseases, lung function, and detection during either wakefulness or sleep. One of the challenges and limitations of TBS recording is the background noise, including speech sound, movement, and even non-tracheal breathing sounds propagating in the air. The breathing sounds captured from the nose or mouth can interfere with the tracheal breathing sounds, making it difficult to isolate the sounds necessary for accurate diagnostics. In this study, two surface microphones are proposed to accurately record TBS acquired solely from the trachea. The frequency response of each microphone is compared with a reference microphone. Additionally, this study evaluates the tracheal and lung breathing sounds of six participants recorded using the proposed microphones versus a commercial omnidirectional microphone, both in environments with and without background white noise. The proposed microphones demonstrated reduced susceptibility to background noise particularly in the frequency ranges (1800–2199) Hz and (2200–2599) Hz with maximum deviation of 2 dB and 2.1 dB, respectively, compared to 9 dB observed in the commercial microphone. The findings of this study have potential implications for improving the accuracy and reliability of respiratory diagnostics in clinical practice.

## 1. Introduction

Tracheal breathing sounds (TBS) have always been beneficial in respiratory assessment, offering non-invasive insights into the functioning of the respiratory system [1,2,3,4]. Recorded at the suprasternal notch, TBS are beneficial in diagnosing and screening a range of respiratory conditions, including obstructive sleep apnea (OSA), asthma, and respiratory infections. The quality of TBS signals plays a vital role in the accurate analysis and interpretation of the diagnosis. However, the accuracy of TBS recordings is often compromised by the presence of background noise—encompassing ambient sounds, patient movement, and the interference of other signals, such as speech and mouth/nose breathing, which can contaminate crucial acoustic details necessary for precise diagnostics. In this paper, we introduce a novel microphone design specifically engineered to capture TBS with exceptional clarity while effectively minimizing background noise interference.

Previous studies have mostly relied on omnidirectional condenser microphones [2,3,5,6]. While effective in capturing sounds from all directions, these devices do not discriminate between TBS and inessential ambient noise. This results in recordings covered with unwanted acoustic signals, requiring sophisticated and computationally expensive post-processing techniques to extract usable diagnostic data. For instance, when recording lung sounds, the use of an air-coupled omnidirectional microphone can result in the desired signal being masked by the ambient breathing sounds captured from the nose or mouth.

The challenge posed by background noise in the recording of TBS has been recognized in clinical practice and research for decades [7]. Efforts to address this issue have included the use of noise-reducing algorithms and advanced signal-processing techniques. However, these methods are time-consuming and can be limited by their complexity and the potential to inadvertently remove clinically relevant sounds along with the noise. Consequently, there remains a substantial need for a more targeted approach to capturing TBS that inherently minimizes capturing unwanted noises at the source.

This study proposes the design of a surface contact microphone designed specifically for the recording of TBS and lung sounds, which incorporates acoustic engineering principles to effectively eliminate the impact of background noise. The new design leverages the use of the piezo-electricity effect to convert the skin displacement due to breathing flow into an electrical signal that could be interpreted as sound.

In summary, the introduction of a noise-eliminating microphone for TBS recording marks a significant advancement in respiratory diagnostics. By addressing the longstanding challenge of background noise in TBS recordings, this technology sets a new standard for the accuracy and reliability of respiratory assessments, potentially transforming both clinical practices and patient outcomes in the field of respiratory care.

## 2. Materials and Methods

### 2.1. Microphone Design and Performance Analysis

Two surface contact microphones were designed for this study with the goal of enhancing the recording quality of tracheal and lung sounds for respiratory screening and diagnosis. The primary focus was to develop microphones capable of detecting tissue displacement caused by airflow turbulence, thereby minimizing the capturing of environmental noise. The first microphone, designed as cost cost-effective solution, comprises a 20 mm in diameter piezo-electric disc paired with JFET circuit due to its low noise characteristics. The size of the piezo disc was selected to achieve an optimal balance between sensitivity and practical application. Smaller discs have lower sensitivity due to their limited surface area, while larger discs are usually less comfortable, and do not fit well at the suprasternal notch, which is the preferred location for TBS recording. A 20 mm disc was determined to be the optimal size. The second design consists of a variable gain amplifier connected to a commercial contact microphone CM-01B. Two tests were performed to evaluate the performance of the microphones against reference microphones.

### 2.2. Test 1: Frequency Response Analysis

The objective of the first test is to compare the microphones’ recordings of white noise with those captured by Brüel and Kjær (B and K) surface microphone type 4948-B, Nærum, Denmark as a reference microphone. The reference microphone was selected due to its superior performance compared to other surface microphones available in the market. Although its frequency response is well-documented in existing data sheets, conducting the test under identical conditions is crucial to ensure a valid comparison. By controlling all variables—such as the speaker type, anechoic chamber, and microphone positioning—the test isolates the microphones’ performance, allowing for a precise evaluation of the new design. The three microphones were placed inside an anechoic chamber at 2 mm distance from a speaker (SP-3114Y Soberton, Minneapolis, MN, USA), as shown in Figure 1.

White noise signal in the frequency range of (20–10,000) Hz was generated for a period of 10 s. The sampling frequency was set at 44,100 Hz for all three microphones. The two proposed microphones were connected through TRS jack to a laptop, while B and K microphone was connected to Edirol R-44 audio recorder. The recordings were conducted sequentially to prevent any interference.

### 2.3. Test 2: Tracheal and Lung Sounds Recording Analysis

A second test was conducted with the purpose of comparing the efficacy of the proposed surface microphones in recording tracheal and lung sounds against Sony ECM-77B, Tokyo, Japan omnidirectional microphone. This microphone was being used since it has been employed in previous studies focused on analyzing tracheal breathing sounds [2,5,8,9]. This comparison allows for assessing the performance of the proposed surface microphones for their intended application, providing a direct benchmark against a microphone commonly used in similar research. The specifications of the two proposed microphones and the reference microphones are presented in Table 1.

Data from six healthy individuals (4 males and 2 females) were included in this test with an average age of 33.6 years and an average BMI of 25.7 kg/m^2^. The participants were instructed to take a deep breath from the nose while keeping the mouth closed and to hold it for 5 s, then exhale. This was then followed by five deep breaths and five tidal breaths which were used for analysis. The microphone in test was placed on the suprasternal notch of each participant for tracheal breathing sounds and on the right side of the chest midclavicular 3rd intercostal space [10] for lung sounds. All participants were sitting in upright position. This test was conducted twice, first in a quiet environment and second with white noise covering a bandwidth of (20–10,000) Hz generated in the background. The frequency range of interest was (100–3000) Hz for TBS and (100–1000) Hz for lung sounds [11]. The sampling frequency for this test was set at 11,025 Hz. The subjects’ position and microphone placement are illustrated in Figure 2.

### 2.4. Pre-Processing and Signal Analysis

All sound signals were band pass filtered (75–3000) Hz using Butterworth filter of order 4 to reduce the effects of heart beats and 60 Hz harmonics. The sound signals were normalized by their variance and standard deviation, and the power spectrum was calculated using Welch method with 10 ms Hanning window size with 50% overlapping between consecutive windows. As for the second test, the central 50% of the time duration around the peak of each respiratory phase was analyzed separately. This duration was selected since the peak of TBS in each phase corresponds roughly to when the breathing airflow plateaus, and hence the signal can be considered stationary [12]. Each breathing phase was normalized by its variance envelope, followed by its standard deviation [13], then segmented using a Hanning window of size 23 ms with 50% overlapping between consecutive segments to calculate the power spectral density (PSD). The average of each breathing phase PSD was then averaged over the number of phases (10 for each respiratory phase). Further details on signal analysis are found in [2]. Moreover, the spectrograms of tracheal and Lung breathing sounds were plotted to visualize the details of the respiratory phases. These visualizations help identify the frequency components impacted by the background noise for each microphone, providing a comparison of their performance in noisy environments. This analysis aids in understanding how effectively each microphone captures the essential sound signals while minimizing noise interference.

### 2.5. Statistical Analysis

A statistical comparison between the two scenarios (without and with background noise) was conducted, denoted by W_1_ and W_2_, respectively. The PSD was calculated for each respiratory phase as outlined in Section 2.2. The frequency range of 200–3000 Hz was divided into seven bands, each 400 Hz with no overlap. The average PSD of each respiration phase of each breath in the dB scale was computed for each frequency band. For example, W_1_–S_2_I_4_–F_3_ corresponds to the average PSD of the 4th inspiration phase (I_4_) of the 2nd participant (S_2_) in the 3rd frequency band (800–1200) (F_3_) Hz for the scenario without background noise (W_1_). In total, each scenario had 30 data points at each frequency band for each respiratory phase. Since the normality test indicated that the data in most frequency bands did not follow a normal distribution, a Wilcoxon test was performed to compare the PSD values between the two scenarios W_1_ and W_2_ at each frequency band (F_k_).

## 3. Results

The average normalized PSD of the microphones’ responses to white noise signal recordings of the first test is shown in Figure 3. The Brüel and Kjær microphone exhibited a perfectly linear response within the tested range of (300–3000) Hz with the smallest RMSE (4.4 dB). This range was chosen because the speaker used in the test has had a linear range starting above 300 Hz, and the upper limit represents the maximum frequency of interest for tracheal breathing sounds. Both the piezo and CM-01B microphones demonstrated an approximately acceptable linear response; however, the piezo microphone showed a lower RMSE (5.8 dB).

Next, the average normalized PSD of the six participants was plotted for each microphone to compare the frequency response with and without the background white noise. Each respiratory phase was plotted separately. Figure 4 shows the effect of the background noise on the Sony microphone for both the inspiration and expiration phases. There is a perfect agreement between both scenarios for the low-frequency range below 1500 Hz. However, the effect of the background noise is clearly noticeable in the high-frequency range (>1500 Hz), especially for the inspiration phase with a maximum deviation of 9 dB between W_1_ and W_2_. Higher amplitudes were observed for scenario W_2_; in addition, multiple resonating frequencies were captured by the microphone. This observation is clearly depicted in the spectrogram of a participant’s TBS where the high-frequency zone is masked out by the background noise (Figure 5). Furthermore, a good agreement between W_1_ and W_2_ scenarios can be observed for both CM-01B and piezo microphones, with the piezo microphone showing better performance (Figures 6 and 8).

Figure 6 and Figure 7 illustrate the average normalized PSD and spectrograms for the CM-01B microphone. The CM-01B microphone shows a consistent response across both scenarios, although background noise slightly impacts the high-frequency range during the inspiration phases. The maximum deviation was found to be 2.7 dB at 2226 Hz. Figure 8 and Figure 9 present the results for the piezo microphone. The piezo microphone maintains a strong performance, with the minimum difference (maximum deviation = 2 dB) observed between scenarios W_1_ and W_2_, indicating its robustness against background noise. Lastly, Figure 10 shows the spectrograms of lung sounds from a participant using the three microphones, providing a visual comparison of their performance with and without background noise. The Sony microphone failed to capture any respiratory phase, while the two proposed microphones, CM-01B and piezo, were clearly able to differentiate the respiratory phases in both scenarios. The piezo microphone demonstrated strong performance with minimal impact from background noise, capturing clear and distinct respiratory sounds. The CM-01B microphone, while showing some susceptibility to movement artifacts, still provided a reliable representation of the lung sounds. These findings highlight the potential of the CM-01B and piezo microphones for accurate respiratory sound measurement in noisy environments.

Moreover, the results of the statistical Wilcoxon test were in line with the previous findings. Table 2 shows the *p*-value at each frequency range for the inspiration phase, indicating the statistical significance of the differences between the scenarios with and without background noise for each microphone. For the Sony microphone, significant differences were observed in the 1800–2199 Hz and 2200–2599 Hz ranges, with *p*-values of 0.002 and 0.03, respectively. This indicates that background noise significantly affects the Sony microphone’s performance in these frequency ranges.

The CM-01B microphone showed a significant difference in the 2200–2599 Hz range, with a *p*-value of 0.01, suggesting some susceptibility to background noise at this frequency range during the inspiration phase. The piezo microphone demonstrated no significant differences across all frequency ranges, with *p*-values well above the threshold for statistical significance. This further supports the piezo microphone’s robustness against background noise. Figure 11 illustrates box plots depicting the distribution of data points from two scenarios, W_1_ and W_2_, across two specific frequency ranges: 1800–2199 Hz and 2200–2599 Hz. The Sony microphone exhibited notably wider interquartile ranges and higher median PSD values in the presence of noise compared to the no-noise scenario across both frequency ranges (1800–2199 Hz and 2200–2599 Hz). This suggests that background noise had a pronounced effect on the measurements captured by the Sony microphone, influencing its ability to accurately detect and differentiate signal strengths in these ranges.

Conversely, the CM-01B microphone showed a mixed result. While there was a significant *p*-value of less than 0.05 indicating differences between scenarios in one frequency range, the presence of outliers suggests variability or potential noise influence in specific measurements rather than a consistent pattern across both ranges. In contrast, the piezo microphone did not show a significant difference between the no-noise and noise scenarios, as indicated by *p*-values greater than 0.05 for both frequency ranges. This suggests that the piezo microphone’s performance in capturing PSD values was less affected by the presence of background noise compared to the other microphones tested.

The statistical results corroborate the earlier observations: the Sony microphone is significantly affected by background noise at higher frequencies, the CM-01B microphone shows moderate susceptibility, mostly due to movement artifacts, and the piezo microphone maintains consistent performance across all tested frequency ranges. These results reinforce the conclusion that the piezo microphone is the most reliable option for capturing respiratory sounds in the presence of background noise.

## 4. Discussion

Acoustic measurements are crucial in numerous scientific, industrial, and medical applications, relying heavily on the accuracy and reliability of the microphone’s performance. The precision of these measurements is significantly influenced by background noise, which can distort acoustic measurements across various microphone types. Microphones exhibit diverse designs and operational principles, leading to differential sensitivity to noise. Condenser microphones, known for their high sensitivity and broad frequency response, are commonly utilized in applications requiring detailed acoustic analysis. Condenser microphones have continuously been used in tracheal breathing sound recordings. Generally, when recording TBS, condenser microphones are placed inside a chamber that is attached to human skin. Breathing airflow inside the airway produces skin vibration, which propagates as sound pressure in the chamber and is subsequently captured by the microphone. This air-coupling mechanism does not prevent the capturing of background noise along with the required sound signal. In addition, the accuracy of the recorded sound signal usually depends on the microphone chamber design [14].

On the other hand, piezo-electric microphones operate on a different principle, converting mechanical vibrations into electrical signals. Their insensitivity to airborne noise and their frequency range make them suitable for medical applications such as respiratory sounds.

Previous studies have discussed the use of contact microphones [15,16,17,18] and accelerometers [19] for recording respiratory sounds. In this study, we analyzed the performance of a newly designed piezo-electric microphone in recording tracheal and lung breathing sounds. Although the primary focus of the design is on recording tracheal breath sounds (TBS), lung sounds were also analyzed to assess the robustness of the microphone. The proposed microphone showed a flat frequency response with a small error difference in addition to its excellent performance when rejecting background noise. The proposed piezo microphone also offers the advantage of being cost-effective, when used as a diagnostic tool, being 500 times cheaper than the B and K type 4948-B microphone and 25 times cheaper than the Sony ECM-77B microphone.

A comparison between the proposed microphone and a condenser microphone (Sony ECM-77B) in recording tracheal breathing sounds has been conducted. Empirical studies highlight the sensitivity of condenser microphones like the Sony microphone to background noise. Noise presence resulted in wider interquartile ranges and higher median PSD values, indicating amplification of both signal and noise components by this type of microphone. However, the proposed microphone maintained consistent performance in noisy environments in all frequency ranges of interest.

The small sample size of this study is a limitation. However, the rigorous data analysis and statistical methods employed mitigate potential biases and provide meaningful insights into the microphones’ performances. Future studies with larger sample sizes could further validate these findings. Another limitation involves movement artifacts when using contact microphones. Although contact microphones are designed to detect vibrations directly from the skin, they are still susceptible to noise introduced by body movements or shifts in microphone placement. This is particularly challenging in clinical settings where patients may not remain completely still. Further research in the microphone chamber design and microphone attachment methods should address this issue for improved performance.

## 5. Conclusions

This study reports the design and testing of a piezo-electric microphone in the application of tracheal breathing sound measurement in comparison to a commercial omnidirectional condenser microphone. Overall, the proposed microphones demonstrated superior performance in recording tracheal and lung breathing sounds in a noisy environment, specifically when capturing high-frequency components. The robust noise rejection capabilities of the piezo-electric microphone suggest its potential utility across various medical applications. These findings highlight the potential of the proposed microphones in enhancing diagnostic accuracy in respiratory medicine, offering reliable detection of respiratory abnormalities even under challenging acoustic conditions. Future research should focus on optimizing the microphone’s chamber design to minimize movement artifacts and further explore its capabilities in clinical settings.

## Figures and Tables

**Figure 1 sensors-24-05511-f001:**
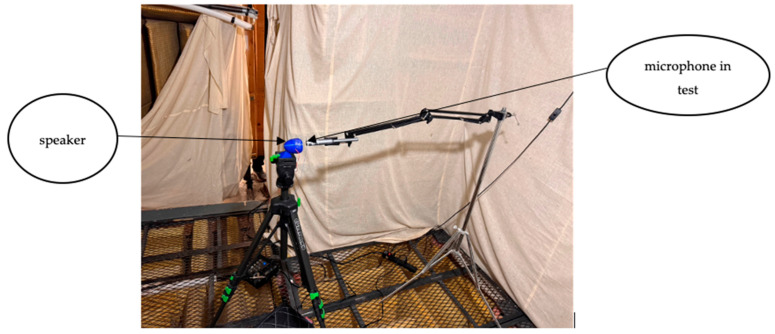
Test 1 setup in the anechoic chamber.

**Figure 2 sensors-24-05511-f002:**
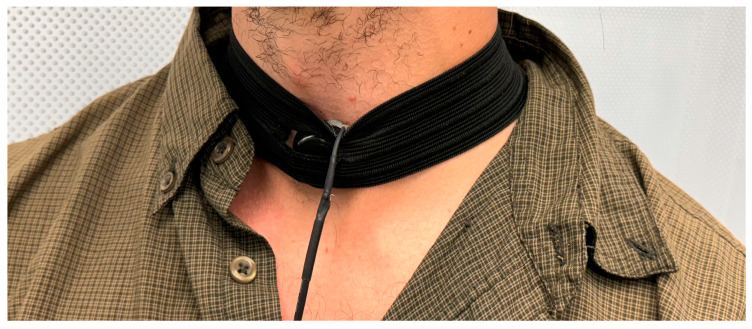
Subject position and microphone placement for tracheal breathing sounds recording (Test 2).

**Figure 3 sensors-24-05511-f003:**
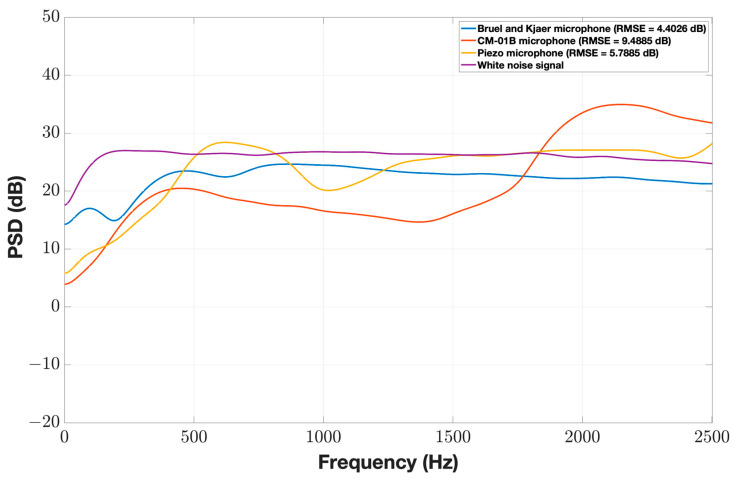
Normalized PSD of the three microphones’ (Brüel and Kjær, CM-01B, and piezo) recordings of white noise signal (Test 1).

**Figure 4 sensors-24-05511-f004:**
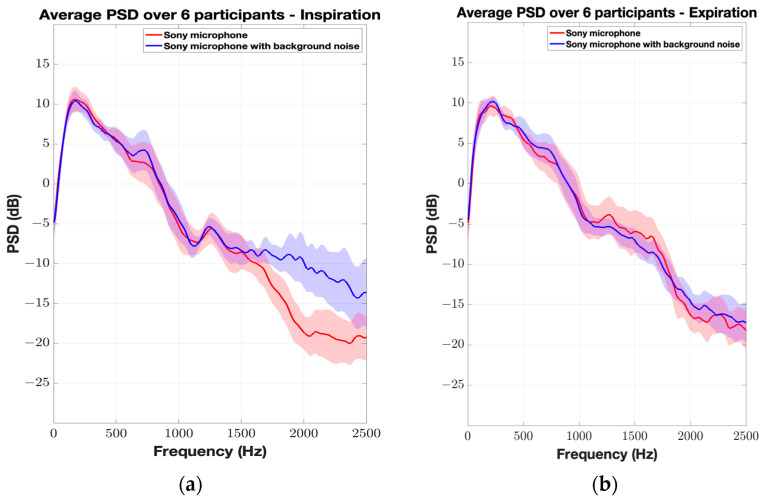
Average normalized PSD of TBS from 6 participants using Sony microphone. The shaded envelope represents the standard error, reflecting variability between participants. (**a**) Inspiration; (**b**) expiration.

**Figure 5 sensors-24-05511-f005:**
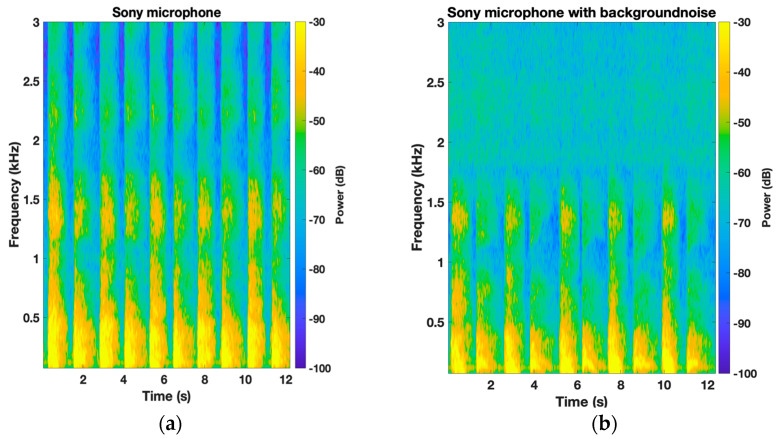
Spectrogram of TBS of a participant using Sony microphone: (**a**) without background noise; (**b**) with white noise in the background.

**Figure 6 sensors-24-05511-f006:**
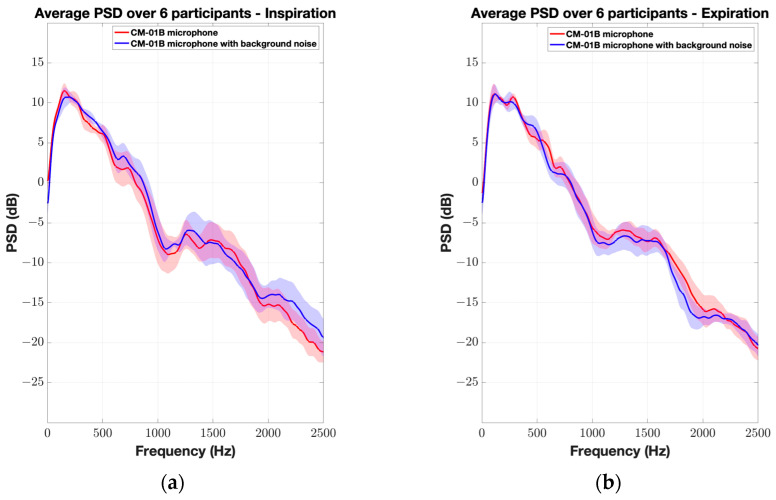
Average normalized PSD of TBS from 6 participants using CM-01B microphone. The shaded envelope represents the standard error, reflecting variability between participants. (**a**) Inspiration; (**b**) expiration.

**Figure 7 sensors-24-05511-f007:**
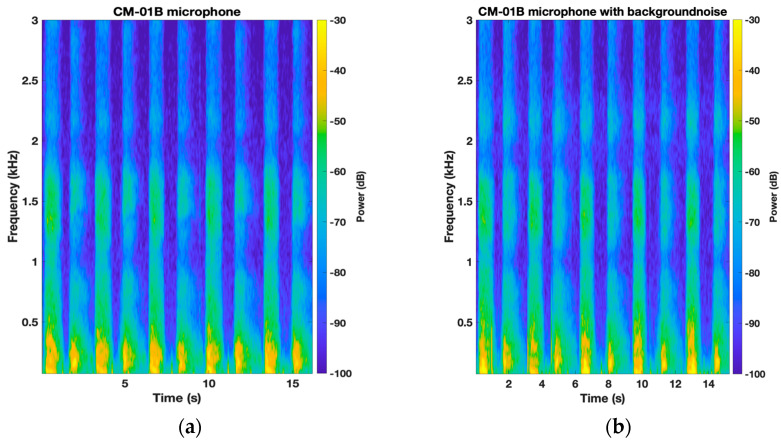
Spectrogram of TBS of a participant using CM-01B microphone: (**a**) without background noise; (**b**) with white noise in the background.

**Figure 8 sensors-24-05511-f008:**
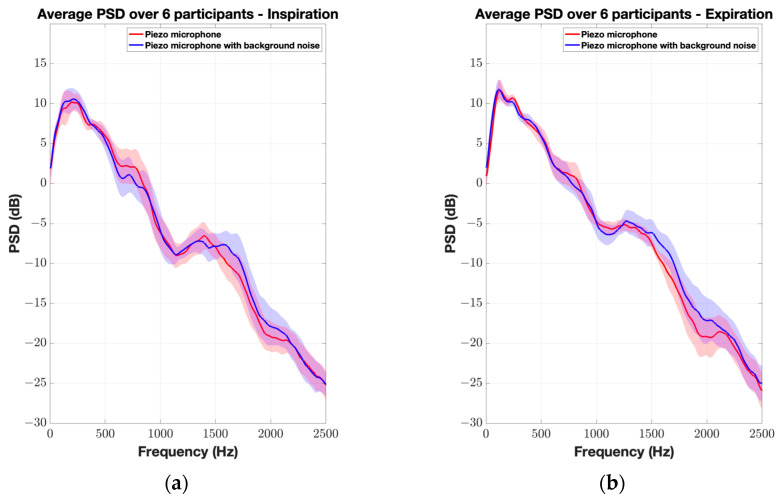
Average normalized PSD of TBS from 6 participants using piezo microphone. The shaded envelope represents the standard error, reflecting variability between participants. (**a**) Inspiration; (**b**) expiration.

**Figure 9 sensors-24-05511-f009:**
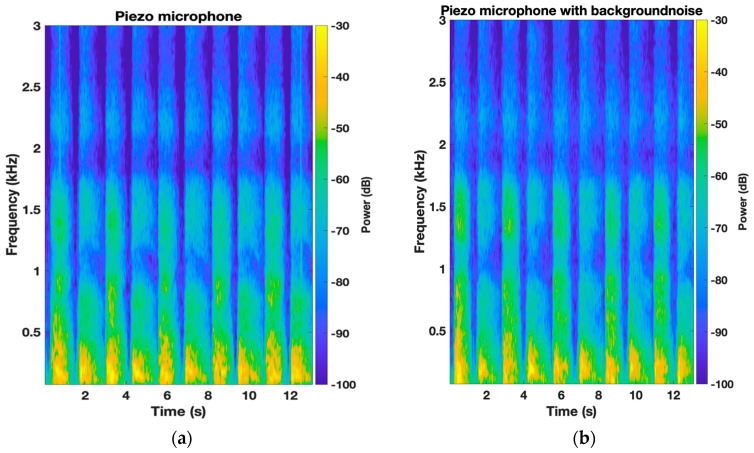
Spectrogram of TBS of a participant using piezo microphone: (**a**) without background noise; (**b**) with white noise in the background.

**Figure 10 sensors-24-05511-f010:**
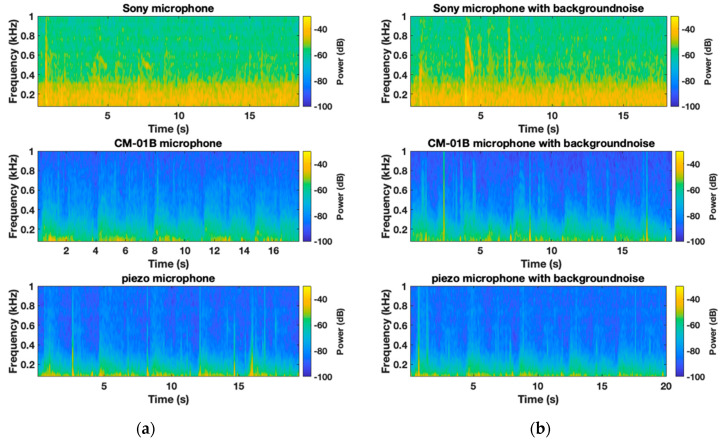
Spectrogram of lung sounds of a participant using the three microphones (**a**) without background noise; (**b**) with white noise in the background; from top to bottom: Sony microphone, CM-01B microphone, piezo microphone.

**Figure 11 sensors-24-05511-f011:**
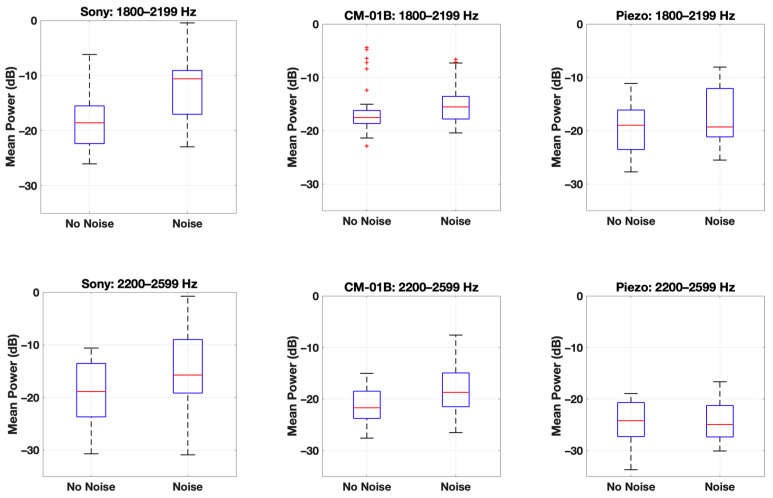
Box plot for the data points of each scenario (W_1_ and W_2_) at two frequency ranges (**top**—1800–2199 Hz, **bottom**—2200–2599 Hz) for the three microphones (**Left**—Sony microphone, **middle**—CM-01B microphone, **right**—piezo microphone).

**Table 1 sensors-24-05511-t001:** The specifications of the microphones used for both tests.

	Sensitivity (Measured at 1 kHz-94 dB SPL)	Frequency Range (Linear)	Sensor Type (Test Involved)
Piezo-electric microphone	−50 dB (3.2 mv/Pa)	(20–5000) Hz	Piezo-electric disc (Tests 1 and 2)
CM-01B	−60 dB (0.9 mv/Pa)	(8–2500) Hz	Piezo-electric (Tests 1 and 2)
Brüel and Kjær type 4948-B	−57 dB (1.4 mv/Pa)	(5–20,000) Hz	Surface microphone (Test 1)
Sony ECM-77B	−52 dB (2 mv/Pa)	(40–20,000) Hz	Condenser microphone (Test 2)

**Table 2 sensors-24-05511-t002:** Absolute mean PSD difference in dB ± standard deviation (*p*-value) in each frequency range for inspiration phase.

	200–599 Hz	600–999 Hz	1000–1399 Hz	1400–1799 Hz	1800–2199 Hz	2200–2599 Hz	2600–3000 Hz
Sony microphone	0.33 ± 1.43 dB (0.67)	0.42 ± 1.74 dB (0.54)	0.46 ± 1.91 dB (0.81)	0.18 ± 3.63 dB (0.68)	4.80 ± 6.82 dB (0.006) *	4.07 ± 6.68 dB (0.03) *	2.80 ± 5.7 dB (0.24)
CM-01B microphone	0.12 ± 1.63 dB (0.69)	1.07 ± 2.15 dB (0.26)	1.03 ± 2.81 dB (0.32)	0.28 ± 2.52 dB (0.82)	1.15 ± 3.42 dB (0.11)	2.81 ± 4.32 dB (0.02) *	1.42 ± 4.50 dB (0.33)
Piezo microphone	0.10 ± 1.19 dB (0.85)	0.81 ± 2.36 dB (0.73)	0.23 ± 2.95 dB (0.78)	0.80 ± 3.20 dB (0.53)	1.5 ± 3.08 dB (0.36)	0.04 ± 2.10 dB (0.71)	0.17 ± 2.19 dB (0.87)

* Statistically significant results (*p* < 0.05).

## Data Availability

Data will be made available upon request to the corresponding author.

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
