# Peer review of "Design and Analysis of a Contact Piezo Microphone for Recording Tracheal Breathing Sounds"

_sensors, 2024, doi:10.3390/s24175511_

Round 1

Reviewer 1 Report

Comments and Suggestions for Authors

Abstract: 

1- The rationale does not precisely resonate with the solution. It is claimed that the mouth/nose sounds interfere with the tracheal sounds. However, the nature of these sounds is related to breathing. Please make it clear what other type of information is needed from tracheal sounds rather than breathing to improve diagnosis. 

2- Please provide some quantitative results in the abstract.

Main text: 

3- "The challenge posed by background noise in the recording of TBS has been recognized in clinical practice and research for decades", please provide appropriate referencing. 

4- Please provide reasons for selecting the size of piezo, type of circuits and microphone and the specific reference microphone. 

5- "50% duration around the maximum of each 105 breathing phase was analyzed separately" Duration of what? If this was the second test, what was the first test. Please report the average duration of the segment to verify the characteristics of the following PSD analysis. 

6- Please re-write the method part, separate different analyses and tests with appropriate subsections and revise the text to solve typos. It would be better to define a name for first and second test to help readers remember what the goal of each was. 

7- What was the reason to use two reference microphones?

8- For figure 3, it would be great to show the variance over participants with shaded envelopes. 

9- What was the bandwidth for the background white noise in the second test? This is important to report to interpret figure 4.

10- In the intro, other disturbances were mentioned such as ambient noises and the sound of breathing collected from nose/ mouth. However, the results were all based on white noise. Please revise the intro based on the results or the second test could be repeated for other noises mimicking different ambient noises or nose/mouth breathing sounds. 

Comments on the Quality of English Language

There are a few grammar problems and typos. 

Reviewer 2 Report

Comments and Suggestions for Authors

Major Comments

·      An overall concern I have with the paper is to a degree its purpose. Test 1- microphone data sheets are already available and evaluate I believe more comprehensively the linear frequency response. What does your test show that is new or different? Test 2- Omnidirectional microphones, in their design, pick up sounds from multiple directions making them prone to background noise, whereas piezo microphones pick up vibrations making them less prone to background noise. Hence the conclusion of the effect of background noise effect appears to be as expected?

·      Paper is hard to follow at the moment. Section 2.1 talks about two tests ending with a brief description of test 2 that becomes the start of Section 2.2. Then test 2 discusses a lung and tracheal sound analysis which appear to be two separate analyses. The results section has a lot of content more suitable for the discussion section. The discussion section has a fair amount of references and information more suitable for the introduction/background section. While in the earlier sections, there is talk of two proposed microphones, in the discussion and conclusion it moves to just a discussion of one proposed microphone. I would recommend making subheadings that highlight Test 1 and Test 2 (or having dedicated sections called Test 1 and Test 2) and properly explain the methodology and setup of these two tests before talking about pre and post-processing.

·      More background information is required about the choice of the microphones and justification of their advantages compared to existing works. In relation to this, highlighting the frequency ranges of tracheal, lung and various noise sounds would be helpful to justify the frequency ranges used for white noise, plots and frequency ranges of the microphones chosen

·      More preamble is required to help explain the purpose and layout of the two (in some ways 3) tests that were performed and their individual aims. For example, it should be mentioned earlier that there is with and without background noise testing for test 2 and clarification on the type and frequency range of that noise.

·      Why did the reference microphone change between the two tests?

·      The results section should be better quantified. For instance, “perfect linear response”- how do you measure perfect? What is an acceptable linear response? How do you quantify perfect agreement?

·      Paper focuses on tracheal sounds, but it at the moment feels a bit jumbled as test 1 is not framed in relation to that, and test 2 also includes lung sound analysis (which is not reflected in the title of the paper, or depth of analysis compared to tracheal sounds).

Minor Comments

·      Summary of key findings should be mentioned in abstract

·      Page 2 paragraph starting line 44 requires appropriate references

· Low cost is mentioned as an advantage a couple of times. This should be quantified in some way. As in, how is this particular microphone more low-cost compared to the other microphones?

·      A figure showing the subject in the appropriate position and placement of microphones would be beneficial for test 2

·      What is the demographic summary of the 6 participants?

·      As sampling frequencies differ from test 1 and test 2, did all microphone circuits have an appropriate anti-aliasing filter?

·      Results section would benefit from figures showing mean (solid line) and standard deviation (shading) db difference of with and without noise on the y-axis and x-axis showing frequency. Similarly, table 2 would benefit from mean and standard deviation differences in each frequency bin, not just p-values.

·      How does this overall study compare with using a background reference microphone and removing sounds that way? What are the benefits of your approach?

·      You mention susceptibility to movement artifacts, further discussion about this would be beneficial. 

·      Figure 9 highlights already that the reference microphone was problematic regardless of whether background noise was present or not. Which appears to make it hard to highlight the effect of background noise interference.

Comments on the Quality of English Language

- Some capitalisation mistakes

- Strong qualitative terms are used, it would be better to quantify before using these terms

Round 2

Reviewer 1 Report

Comments and Suggestions for Authors

Thanks for the revisions!

Comments on the Quality of English Language

The flow of the sentences and the use of words can be improved. 

Reviewer 2 Report

Comments and Suggestions for Authors

My comments have been sufficiently addressed. 

Comments on the Quality of English Language

Minor proofreading is required